# SMART System in the Assessment of Exercise Tolerance in Adults

**DOI:** 10.3390/s23249624

**Published:** 2023-12-05

**Authors:** Katarzyna Nierwińska, Andrzej Myśliwiec, Anna Konarska-Rawluk, Anna Lipowicz, Andrzej Małecki, Andrzej Knapik

**Affiliations:** 1Institute of Physiotherapy and Health Sciences, Academy of Physical Education, 40-065 Katowice, Poland; k.nierwinska@awf.katowice.pl (K.N.); nimerin@gmail.com (A.K.-R.); anna.lipowicz@upwr.edu.pl (A.L.); a.malecki@awf.katowice.pl (A.M.); aknapik@sum.edu.pl (A.K.); 2Department of Antropology, Wrocław University of Environmental and Life Sciences, 50-375 Wrocław, Poland; 3Department of Adapted Physical Activity and Sport, School of Health Sciences in Katowice, Medical University of Silesia in Katowice, 40-752 Katowice, Poland

**Keywords:** physical-activity, 6MWT, SMART system

## Abstract

Health-oriented physical activity should meet two key criteria: safety and an optimal level of exercise. The system of monitoring and rationalization of training (SMART) was designed to meet them. SMART integrates a custom-configured inertial measurement unit (IMU) and a sensor with real-time heart rate measurement (HR) using a proprietary computer application. SMART was used to evaluate the safety and exercise load with 115 study participants: 51 women (44.35%) and 64 men (55.65%) aged 19 to 65 years. The exercise test was the 6MWT test. In 35% of the participants, the mean HR exceeded the recognized safe limit of HR 75% max. Ongoing monitoring of HR allows for optimal exercise and its safety. Step count data were collected from the SMART system. The average step length was calculated by dividing the distance by the number of steps. The aim of the present study was to assess the risk of excessive cardiovascular stress during the 6MWT test using the SMART system.

## 1. Introduction

The optimal healthy level of physical activity (PA) has been repeatedly explored and confirmed [1,2,3]. It provides many benefits, such as improving the function of many body organs and systems [4,5,6,7,8,9,10,11], reducing overall mortality in the population, and, most importantly, significantly reducing the risk of death from cardiovascular disease [12]. Some researchers have shown that people who are physically active compared to those with a sedentary lifestyle can live 7 to 8 years longer [13,14].

Despite the significant health benefits of regular PA, intense exercise can paradoxically act as a trigger of life-threatening ventricular arrhythmias (VA) in the presence of CVD. In fact, sudden cardiac death (SCD) is the leading cause of exercise-related mortality among young athletes [15,16,17].

Key to the health benefits is the optimal level of PA. Physical exercise undertaken inappropriately, too intensely, for too long a duration, without prior preparation, and without taking into consideration comorbidities can pose a real threat to health and life [18]. Among the serious cardiovascular events associated with physical activity are sudden cardiac arrests, sudden cardiac death, acute coronary syndromes, transient ischemic attacks, or supraventricular arrhythmias [19]. The etiological factor in the occurrence of these events is recognized or unrecognized cardiovascular disease or an inappropriate form of exercise. A higher risk has been found in men (3–9 times higher than in women) [20], in black athletes [21], and in some team sports, e.g., basketball or soccer [22].

Both the European Society of Cardiology and the 2018 Physical Activity Guidelines (PAG) indicate that moderate-intensity physical activity should be the goal for people with low and average fitness. Moderate intensity exercise, as defined by the American College of Sports Medicine (ACSM), is exercise at 64–76% of your maximum heart rate. Following the cited recommendations, we adopted a value of 75% HRmax as the boundary between moderate and high exercise intensity. Our study focused on people practicing recreational sports. In this population—unlike professional athletes—risk factors for atherosclerosis and diagnosed CVD are more common. High exercise intensity increases the risk of adverse cardiac events [23].

However, regardless of the risks involved, which are primarily related to intensive forms of PA, there is a general consensus among researchers on the beneficial effects of activity on cardiovascular fitness who suggest studying the dose–response relationship in the context of specific types of PA [24]. To date, guidelines developed in 2020 by the European Society of Cardiology have been commonly used. They systematize knowledge of the duration and intensity of exercise in both healthy adults and in specific populations such as patients with obesity, hypertension, diabetes, and patients over 65 years of age [25]. 

6 minute walk test (6MWT)

The 6-minute walk test is often used to assess functional status both in healthy populations [26,27] and in clinical practice [28,29,30,31,32]. In clinical terms, the 6MWT is mostly used to assess exercise performance in cardiac rehabilitation [28], pulmonology [29], and a number of other fields [30,31,32]. More recently, due to the ongoing pandemic, it has been employed for the diagnosis of asymptomatic exercise-induced hypoxia in patients with COVID-19, or for assessing the fitness of COVID-19 survivors [33]. The consequences of pandemic-related restrictions, such as a reduction in interpersonal contact, cessation of many group activities, the discontinuation of team sports by non-professionals, and the closure of many exercise-related recreational facilities such as parks and swimming pools, have resulted in increased sedentary lifestyles for many people, a decline in overall fitness, and a deterioration in the general health of the population [34]. The problem of declining PA and failure to return to the former level of physical activity particularly affects individuals who have been diagnosed with COVID-19 [35,36,37]. In addition to somatic symptoms, the problem is kinesiophobia [38]. The fear of undesirable consequences of PA may cause people to avoid exercise, especially its more intense forms. The aforementioned rationale suggests undertaking safe forms of PA, such as walking. 

SMART

The safety of walking as a form of PA has long inspired researchers to develop standards (reference values) for the 6MWT in various populations [26,27,39,40,41]. The need for health and the optimization and safety of physical exercise has led to a rapid growth in research on activity-monitoring devices [35]. This motivated the creation of a team of researchers from different areas of science for the development of a system for the monitoring and rationalization of training (SMART). The SMART project is focused on physical activation—ordered, monitored, and controlled by using a personalized algorithm for optimizing safe physical exercise and physioprevention in the field—gender and body composition, long-term performance and physical fitness, supporting decision-making regarding the forms of physical activity undertaken for the behavioral profile and health status, and monitoring the user’s condition in real time. The combination of data obtained from measurements of people from the population of users of similar age and with a similar anthropometric profile will constitute the basis to form a warning against engaging in dangerous behaviors and will also suggest tasks in the field of physioprevention. SMART integrates a custom-configured inertial measurement unit (IMU) and a sensor with real-time heart rate measurement (HR) with a proprietary computer application. The use of the IMU (accelerometer, gyroscope, and magnetometer) in SMART allows for the qualitative measurement and reporting of movement, whereas supplementing the system with HR measurement allows for ongoing monitoring and recording of exercise intensity. HR is measured using the Polar H10 chest strap. The sensor reads heart rate, or more precisely, the electrical discharges of the heart muscle, and records them in real time. The data such as pulse (HR—heart rate), RRS–R-R intervals [1/1024], and RRMS–R-R intervals [ms] are output at a frequency of 1 Hz in the form of a list. Timestamps are assigned by the system that collects and processes the data. This type of measurement is widely considered to be very accurate. 

Study aim

The aim of the present study was to assess the risk of excessive cardiovascular stress during the 6MWT test using SMART. We evaluated the associations of participants’ exercise performance and basic morphological data with the risk of exceeding HR values considered safe during the 6MWT.

## 2. Materials and Methods

We examined 115 individuals: 51 women (44.35%) and 64 men (55.65%) aged 19 to 65 years (mean 37.40; SD = 12.26). The selection for the study was purposive. The participants were volunteers of the SMART research program. Volunteers were recruited using social media and among friends of members of the research team. The following selection criteria were adopted: voluntary participation: confirmed by informed consent to participate in the study;age: 18–65 years;no health contraindications to perform exercise testing. This criterion was verified by conducting a medical examination before the tests included in the SMART program.

All participants were informed in detail about the research process and the possibility of withdrawing from the program at any time without giving reasons. The program was approved by the Bioethics Committee KBE 8/2020. 

Validation of the SMART system heart rate sensor

HR measurement was validated using the cardiopulmonary exercise test (CPET). The validation involved 20 participants and the duration of the test was 10 min. Heart rate at the first second and the following minutes were taken into account. During CEPT, study participants wore a Polar H10 heart rate monitor implemented in the SMART system. The CPET test was performed using the VYNTUS CPX ergospirometer with the VYNTUS ECG module, manufactured by Vyaire (Chicago, IL, USA) The following evaluation algorithm was used: The first second and the following minutes of HR were taken into account. Mean HR was calculated based on SMART (Polar H10 heart rate monitor) and ECG measurements;The difference between SMART and ECG measurements (absolute values) was calculated;The difference between SMART and ECG measurements in % [(difference in absolute values/mean heart rate from SMART and ECG measurements) x 100] was calculated.

The mean absolute difference in HR was 2.24 (SD = 1.97), while in percentage terms, this was a mean of 2.49 (SD = 2.39). As can be seen from the above data, the POLAR H10 heart rate sensor implemented in the SMART system allows measurements with sufficient accuracy.

Stages of the research

Data were collected on the morphological parameters of the participants. Body height and waist and hip circumferences were measured to calculate the weight-to-height ratio (WHR). Body weight, BMI, and %FAT data were obtained from BIA analysis using an In Body 770 device.Aerobic capacity levels were estimated using the Astrand–Rhyming step test. This indirect method of assessing the maximal oxygen uptake uses the established relationship between exercise oxygen uptake and heart rate. The qualitative VO2max assessment was made according to standardized criteria [38]. Next, for statistical analyses, the participants were divided into three groups: I—very low or low exercise performance; II—average or good exercise performance; III—very good or highest exercise performance.The participants performed the 6MWT test. This test was conducted on a running track with a 400 m circumference. The distance covered was measured to the nearest 1 m using a tape measure. Step count data were collected from the SMART system. The mean step length was calculated by dividing the distance by the number of steps.

Statistical analysis

Descriptive statistics (medians and min–max values) were calculated. Nonparametric analyses (Kruskal–Wallis and U Mann–Whitney tests, ANOVA) were used for intergroup comparisons. The statistical significance level was set at *p* < 0.05.

## 3. Results

Based on descriptive statistics and group comparison, the qualitative assessment of VO2max did not reveal differences between the participants, either in terms of age or the morphological parameters studied. 

Intergroup comparisons of walking parameters showed differences only in one case. This concerned the mean HR in women.

According to the aim of the study, we measured HR at individual minutes. We calculated the percentages of the maximum heart rate, and how many people exceeded the safe heart rate threshold set at 75% of the maximum heart rate in the 6MWT test.

We also made a comparisons of the number of people who exceeded the accepted safe heart rate threshold (75% HRmax) during the entire 6MWT—due to the qualitative assessment of VO2max. Among women, the comparison showed differences between groups: chi2 = 11.98, *p* < 0.01. There were no differences in men: chi2 = 3.04, *p* > 0.05—Figure 1.

Both women and men who exceeded our HR limit achieved higher results in the 6MWT distance. The levels of differences were respectively: *p* < 0.05 in women and *p* < 0.01 in men—Figure 2.

## 4. Discussion

The aim of our research was to assess the risk of excessive cardiovascular stress by estimating the level of exercise intensity achieved during the the 6MWT test and monitoring the heart rate tested in real time using the SMART system. Our research clearly shows that the intensity of exercise achieved, especially by people with low and average fitness, even in the 6MWT test, generally considered as a safe, exceeds the limits considered safe.

The qualitative assessment of the results of the 6MWT test is based on the evaluation of walking parameters, primarily distance, thus providing indirect information about the exercise performance of the participants [35,39,40,41,42,43]. A number of factors influence the results of the 6MWT test, mostly the current health status, both physical and mental, as well as motivation. These are the reasons why, according to the recommendations of the American Thoracic Society (ATS) [44], different reference equations are being developed for different healthy populations [40,45,46]. There is also no pattern for the interpretation of the results of the 6MWT tests expressed as percentages of the normal value. Troosters et al. proposed that 6MWT results that are below 82% of the normal value should be considered abnormal [47]. There are also formulas to determine the lower limit of the normal values for the 6MWT [48,49]. It is recommended that the analysis of changes in the distance in the 6MWT test should take into account the data on the minimum important difference (MID) for a given disease entity or clinical situation. The MID is defined as the minimum change in the 6MWT that is considered to be significant and that the person considers an improvement or worsening of functional status [50]. The cut-off values for the MID are highly variable and depend on the type of condition [50,51,52,53]. The choice of normal values depends on the team performing the tests and the reference group, which can affect the reproducibility of test results and their interpretation. 

In a study conducted on healthy adults, significant differences were observed in distance covered [49,51]. The data obtained in our study on the walking parameters such as distance or number of steps were within the ranges of results presented in previous studies [48,54,55]. However, the mean distance values in our study in groups of women and men were significantly higher [48,54,55,56]. The average 6MWT score in our participants was ~700 m and was in the upper ranges of distances reported in other studies (Table 1) [48,54,55,56]. The distance covered by the participants during the 6MWT test did not differ by aerobic capacity. However, it should be noted that the test was performed under relatively optimal conditions on a running track with a 400 m circumference. Another factor was the choice of the participants, who were volunteers, which may have affected the level of motivation during the 6MWT exercise. During our research, some participants, especially younger ones, who are regularly active, found that walking was not intense enough for them.

Heart rates during submaximal efforts are also affected by other factors, such as emotions [57,58]. The effect of emotions on HR during exercise decreases with an increase in their intensity [57]. The heart rate during submaximal exercise is also modified by environmental factors such as the ambient temperature [59]. The repeatability of HR during exercise with identical load is surprisingly high [60]. The standard deviation of HR during repeated standardized exercise is only ±5 beats per minute, provided the test is performed accurately enough. During walking, heart rate shows a negative correlation with other indicators of physical fitness [61]. These comments can be taken as some limitations on the interpretation of the results.

The primary objective of the 6MWT test is to determine the exercise tolerance of the person. According to the publications available, the 6MWT is a relatively safe test [49,56,62]. However, physical exercise is physiological stress and can induce cardiovascular abnormalities in people of all ages regardless of fitness level [62]. The nature of the exercise response (atonic, normotonic, hypertonic, hypotonic) and the time to reach submaximal heart rate (HRsubmax) allows for optimizing the safety of the test during its performance [63,64].

A comparison of age and the morphological parameters studied (BMI, WHR, %FAT) with the estimated level of fitness showed no differences among the research groups (Table 2). Studies on the relationship between exercise performance estimated using indirect methods to assess VO2max and morphological indices are conflicting. The authors have shown an inverse correlation of BMI, WHR, and %FAT with the level of VO2max [65,66]. Lohman et al. (2008) demonstrated an indirect relationship between functional capacity and BMI [67]. In contrast, Gonzales-Suarez et al. (2011) found no relationship between BMI and the level of fitness estimated based on VO2max in children [68]. It seems that these discrepancies and the lack of relationships between the somatic profile of our participants and the results may explain the individually varying levels of volitional traits. 

According to the ACSM recommendations, the exercise intensity in healthy adults estimated based on HR_max_ should range between 65 to 85% of its maximal level [69]. It may be debatable that we arbitrarily set the safe heart rate threshold at 75% of HR_max_. It should be noted that various formulas are proposed to determine the predicted value of HR_max_ [70]. However, researchers agree that heart rate is an indicator of workload, and the easiest indicator of cardiovascular response to exercise to measure. The increase in heart rate occurs almost immediately at the beginning of an exercise. It is characterized by a rapid initial phase, associated with a decrease in the activity of the parasympathetic part of the autonomic system, and a subsequent slower phase that depends on an increase in the activity of sympathetic innervation [70]. It is assumed that the time required to stabilize HR at the level of exercise load increases with an increase in the intensity of work (load). During a submaximal exercise with low intensity, such as the 6MWT, the time to reach functional equilibrium should be 2–3 min [71]. These patterns were observed in our participants.

If we want to apply the above recommendations in practice, it should be assumed that a healthy man aged 45 can without significant risk perform physical exercises that will accelerate the heart rate to 105–130 bpm, while a man aged 65 should not exceed the heart rate of 115 bpm. Of course, threshold values should not be treated arbitrarily as healthy and trained people can tolerate much higher intensity efforts. Our research clearly showed that in groups of subjects whose performance was assessed at a weak or average level, the safe threshold of heart rate, which we assumed at 75% HR_max_, was more often exceeded (Table 3). This was particularly visible in the group of women whose performance was rated as average. Exceeding the threshold value of heart rate also occurred in the group of subjects whose performance was assessed as high, but this happened much less frequently.

There are many recommendations regarding the level of physical activity in the prevention of cardiovascular diseases and health promotion, published and popularized by various scientific societies [72]. In many countries, recommendations made by cardiologists, sports physicians specializing in preventive medicine, and family physicians compete with each other. This variety of recommendations and not always consistent positions, especially when it comes to the intensity or duration of exercise, may undoubtedly confuse not only people interested in taking up physical activity, but also doctors and personal trainers. Monitoring the heart rate in real time allows you to personalize the intensity of physical activity and clearly defines its safety margin.

Generally, various forms of exercise are recommended (walking, jogging, cycling, swimming) and, as a supplement to basic training, resistance exercises, which should constitute approximately 10–15% of the training volume. As your training level increases and your exercise tolerance is good, it is recommended to extend the duration of exercise and increase its intensity [71]. In light of health-promoting recommendations, it seems very important to enable people undertaking physical activity to assess their own performance capabilities. It also seems sensible to disseminate knowledge and tools enabling the determination of a safe range of physical intensity undertaken by potentially healthy people with average and low physical fitness. Research shows that the risk of cardiac events associated with physical activity is relatively low [73,74,75]. Developing good exercise habits and taking care of your cardiovascular system significantly reduces this risk. Factors that increase the risk of cardiovascular events or even death during physical activity include non-compliance with recommendations regarding the intensity and duration of exercise.

## 5. Conclusions

There are a number of arguments for the 6MWT to be used as widely as possible, not only in clinical practice but also in relatively healthy individuals who need to monitor their exercise performance levels. The use of systems for the real-time monitoring of exercise load not only increases the safety of exercisers but also allows the presentation of the rationale for undertaking or continuing activity at a certain level of intensity [72,75]. This is important for taking care of individual health. Our study unequivocally demonstrated the need for the ongoing monitoring of heart rate and the application of detecting the exceeded safe threshold of exercise, especially for individuals with low and average fitness. Intergroup comparisons of the walking parameters showed differences in only one case. This concerned the mean HR for women.

## Figures and Tables

**Figure 1 sensors-23-09624-f001:**
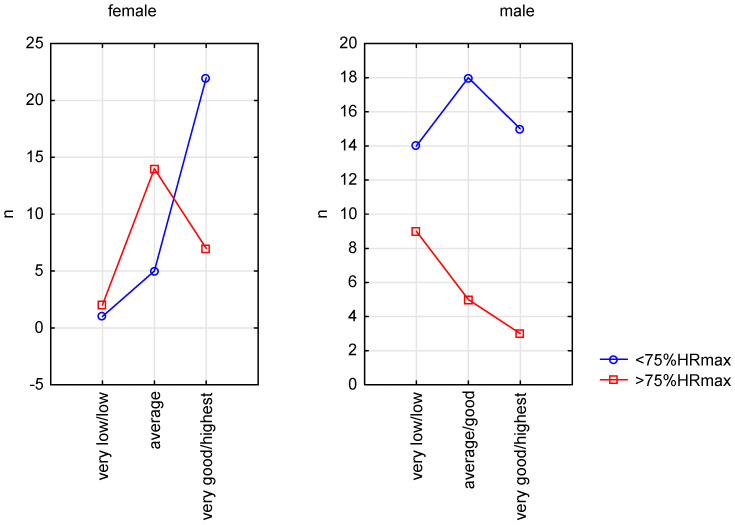
Exceeding 75% HRmax during the 6MWTaccording to performance groups.

**Figure 2 sensors-23-09624-f002:**
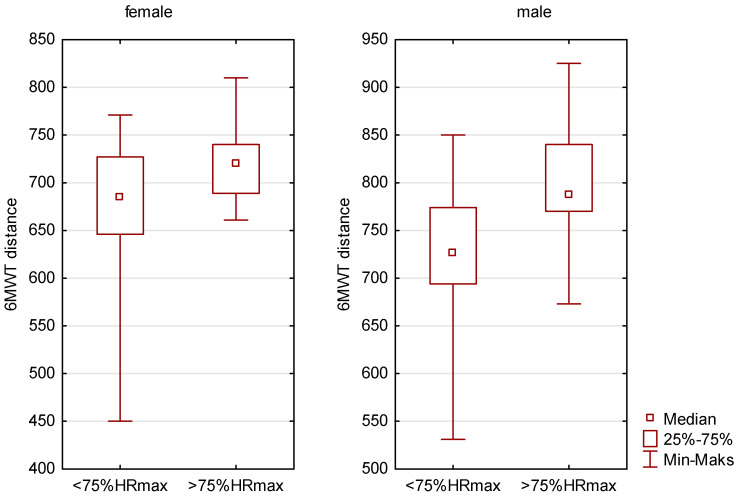
Comparison of the distance of people <75%HRmax with people >75%HRmax.

**Table 1 sensors-23-09624-t001:** Age and morphological parameters of participants by fitness levels.

Sex	Pl	*n*	Age [Years]		Body Height [cm]		Body Weight [kg]		BMI		WHR		%FAT	
			Median Min–Max	*p*	Median Min–Max	*p*	Median Min–Max	*p*	Median Min–Max	*p*	Median Min–Max	*p*	Median Min–Max	*p*
female	I	3	38.00 22–47	>0.05	169.60 169–171	>0.05	62.00 53.20–68.10	>0.05	21.30 20.10–22.70	>0.05	0.76 0.72–0.83	>0.05	25.40 20.40–26.80	>0.05
	II	19	44.00 19–58		164.50 148–175		63.40 40.00–81.70		23.80 16.80–31.90		0.87 0.65–0.94		27.90 17.20–35.30	
	III	29	32.00 20–57		166 148–176		67.10 57.50–78.50		22.80 18.40–28.90		0.79 0.69–0.95		23.50 10.70–34.50	
male	I	23	39.00 19–65	>0.05	177.80 162–187	>0.05	81.80 72.10–95.80	>0.05	24.70 19.10–35.30	>0.05	0.88 0.77–1.02	>0.05	18.00 8.20–30.80	>0.05
	II	23	34.00 20–63		182.30 168–195		81,80 61.00–113.60		24.40 20.20–31.20		0.89 0.78–1.03		18.80 11.00–31.40	
	III	18	38.50 20–65		182.10 162–193		80.10 57.50–113.20		23.75 21.00–28.70		0.88 0.81–1.00		17.00 9.30–23.10	

Notes: Pl—exercise performance level; I—very low, low; II—average, good; III—very good, highest.

**Table 2 sensors-23-09624-t002:** Parameters of the 6MWT test: descriptive statistics.

Sex	Pl	*n*	Distance [m]		Number of Steps		Mean Step Length [m]		Mean HR 1–6′	
			Median Min–Max	*p*	Median Min–Max	*p*	Median Min–Max	*p*	Median Min–Max	*p*
female	I	3	714 667–751	>0.05	819 780–828	>0.05	0.87 0.86–0.91	>0.05	161.00 108.17–173.50	<0.05
	II	19	689 573–771		804 653–909		0.85 0.71–1.12		146.50 73.67–170.83	
	III	29	710 450–810		820 600–960		0.86 0.67–1.16		126.00 96.67–162.00	
male	I	23	731 600–910	>0.05	767 639–902	>0.05	0.93 0.78–1.11	>0.05	131.67 101.83–153.17	>0.05
	II	23	727 531–888		798 724–853		0.92 0.73–1.05		125 96.83–159.17	
	III	18	778.50 625–925		809.50 454–883		0.96 0.87–1.57		118.42 100.33–150.50	

Notes: Pl—exercise performance level; I—very low or low; II—average, good; III—very good, the highest.

**Table 3 sensors-23-09624-t003:** The 6MWT: HR, percentage of HR_max_, and individuals who exceeded 75% of HR_max_ at individual minutes of the test.

Sex	6MWT Min	Group								
		I			II			III		
		HR	% HR_max_	*n* (%) Persons >75% HR_max_	HR	% HR_max_	*n* (%) Persons >75% HR_max_	HR	% HR_max_	*n* (%) Persons >75% HR_max_
female	1′	136	74	2 (67%)	129	72	6 (32%)	124	67	3 (10%)
	2′	144	78	2 (67%)	136	76	14 (74%)	129	70	7 (24%)
	3′	147	80	2 (67%)	139	78	14 (74%)	130	71	7 (24%)
	4′	152	83	2 (67%)	147	83	15 (79%)	131	71	9 (31%)
	5′	153	83	2 (67%)	147	82	15 (79%)	131	71	8 (28%)
	6′	154	84	2 (67%)	145	81	16 (84%)	127	69	5 (17%)
male	1′	125	70	5 (22%)	116	63	0	116	64	0
	2′	131	73	11 (48%)	122	66	5 (22%)	119	66	2 (11%)
	3′	133	74	12 (52%)	125	68	6 (26%)	121	67	3 (17%)
	4′	135	75	11 (48%)	127	69	7 (30%)	124	68	4 (22%)
	5′	136	76	15 (65%)	129	70	5 (22%)	125	69	5 (28%)
	6′	132	73	10 (43%)	127	70	6 (26%)	123	67	5 (28%)

## Data Availability

Institute of Physiotherapy and Health Sciences, Academy of Physical Education in Katowice.

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
