# Peer review of "SMART System in the Assessment of Exercise Tolerance in Adults"

_sensors, 2023, doi:10.3390/s23249624_

Round 1

Reviewer 1 Report

Comments and Suggestions for Authors

Thank you for the opportunity to review this manuscript. I found this paper very interesting as is valuable in terms of the tests it has targeted. There are some issues that need to be addressed in my view.

Abstract

An aim is required.

Introduction

Line 41 can be removed as it does not add anything to the introduction.

Line 64: ‘…undergone COVID-19’. Please replace undergone with ‘been diagnosed with…

The use of 75% of HR maximum must be justified.

Methods

Please provide the body mass and height of the participants if they are available.

Please remove the bullet points and write in paragraph style.

Please provide the manufacturer details of the ECG system.

Please include appropriate effect size calculations.

Results

Please provide the p-values in tables 1 and 2 as well as units of measurement for the appropriate variables.

Discussion

In the first paragraph restate the aim of the study followed by the findings of your study.

The current first paragraph of the discussion appears to be interpretation of the results. I would suggest placing this paragraph pater on in the discussion.

Line 224: Please replace [PMID: 10919953] with a reference.

Lines 228-229: It says no relationship was found between age and morphological parameters and the estimated level of fitness. This cannot be said as there was no statical analysis conducted to determine the relationship. If a relationship is to be included, then the appropriate statistical test needs to be used. Furthermore, this would be an additional aim to the study.

Comments on the Quality of English Language

The quality of English is good but needs a careful read to make the text clearer.

Author Response

Reviewer 1:

General comment: Thank you for the opportunity to review this manuscript. I found this paper very interesting as is valuable in terms of the tests it has targeted. There are some issues that need to be addressed in my view.

Answer: Thank you very much for your comments which have helped us improve this manuscript.

Comment 1: Abstract; an aim is required.

Answer: Thank you for your comment. We agree with you and we have changed it in the article. The research aim has been added to the abstract.

Comment 2: Introduction; line 41 can be removed as it does not add anything to the introduction.

Answer: Line 41 has been removed

Comment 3: Introduction; line 64: ‘…undergone COVID-19’. Please replace undergone with ‘been diagnosed with…

Answer: ‘Undergone COVID-19’ has been replaced by ‘been diagnosed with’

Comment 4: Introduction; the use of 75% of HR maximum must be justified.

Answer: Thank you very much for your attention. Both the European Society of Cardiology and the 2018 Physical Activity Guidelines (PAG) indicate that moderate-intensity physical activity should be the goal for people with low and average fitness. Moderate-intensity exercise, as defined by the American College of Sports Medicine (ACSM), is exercise at 64-76% of your maximum heart rate. Following the cited recommendations, we adopted a value of 75% HRmax as the boundary between moderate and high exercise intensity. Our study focused on people practicing recreational sports. In this population – unlike professional athletes – risk factors for atherosclerosis and diagnosed CVD are more common. High exercise intensity increases the risk of adverse cardiac events. This information has been added to the manuscript.

Comment 5: Methods; please provide the body mass and height of the participants if they are available.

Answer: The required information was included in the text in the methods and results subsections

Comment 6: Methods; please remove the bullet points and write in paragraph style.

Answer: The bullet points have been removed and the paragraph style has been corrected

Comment 7: Methods; please provide the manufacturer details of the ECG system.

Answer: The manufacturer details have been provided.

Comment 8: Methods; please include appropriate effect size calculations.

Answer: Thank you very much for your comment. We assumed that to calculate the effect size, the distributions must be normal. Non-parametric statistics were used here because the assumption of normality of distribution was not met.

Comment 9: Results; please provide the p-values in tables 1 and 2 as well as units of measurement for the appropriate variables.

 Answer: P-values in tables 1 and 2 have been completed

Comment 10: Discussion; in the first paragraph restate the aim of the study followed by the findings of your study.

Answer: We followed your suggestion, thank you for your comment.

Comment 11; Discussion; the current first paragraph of the discussion appears to be interpretation of the results. I would suggest placing this paragraph pater on in the discussion.

Answer: The discussion has been changed according to your suggestions.

Comment 11: Discussion; Line 224: Please replace [PMID: 10919953] with a reference.

Answer: [PMID: 10919953] has been replaced. We are very sorry for our mistake.

Comment 12: Discussion; lines 228-229: It says no relationship was found between age and morphological parameters and the estimated level of fitness. This cannot be said as there was no stoical analysis conducted to determine the relationship. If a relationship is to be included, then the appropriate statistical test needs to be used. Furthermore, this would be an additional aim to the study.

Answer:

Comment on the Quality of English Language: The quality of English is good but needs a careful read to make the text clearer.

Answer: The text has been checked again for linguistic correctness. The corrections have been marked in red.

Reviewer 2 Report

Comments and Suggestions for Authors

This is a very relevant paper in the context of timing and the need for further research as around the world the relevance of exercise-related health concerns is on the rise. People of all age groups including young people suffer from sudden cardiac arrest with excellent training and exercise history. So a need to effectively measure exercise limits is essential. The paper is prevalent in the context of using multiple sources of data measurements including IMU and heart rate measurements to understand unique outputs from different individual groups.

The introduction and entire article have an excellent flow of background context and relevant references. From the technical point of view, the paper provides a straightforward approach and the scientific experiment seemed to be sound in nature using inertial sensors and real-time heart measurement sensor output parameters to make judgements. NASA and other similar organisations have confirmed trampolining and the positive effects of g-force as an excellent form of exercise. It is suggested that the references in line 29 also include the following trampolining reference: Eager D., Chapman C., Bondoc K., Characterisation of trampoline bounce using acceleration, 7th Australasian Congress on Applied Mechanics ACAM7, Adelaide, Australia, 9-12 December 2012.

Sufficient clarification was made on the data collection samples, techniques and devices utilised. The results calculation conditions are clear and straightforward. The discussion has a broader context.

The following specific suggestions on the manuscript are tabled:

A brief background on unexplanatory heart failures in young adults with good exercise history ought to be included in the introduction.

The discussion should refer to the individual figure and table data in the sentences where applicable.

In the abstract please include how the integration of different components in the SMART system contributed to the exercise tolerance assessment besides HR loading (Polar H10 chest strap) and its effectiveness. This can include the execution and protocols used from the SMART system factors/elements to arrive at the results.

In lines 124-125, the point "mean heart rate from SMART and ECG measurements" is ambiguous, please rephrase/elaborate.

Comments on the Quality of English Language

The Patent and funding sections at the rear of the paper require editorial correction.

Author Response

General comment: This is a very relevant paper in the context of timing and the need for further research as around the world the relevance of exercise-related health concerns is on the rise. People of all age groups including young people suffer from sudden cardiac arrest with excellent training and exercise history. So a need to effectively measure exercise limits is essential. The paper is prevalent in the context of using multiple sources of data measurements including IMU and heart rate measurements to understand unique outputs from different individual groups.

Answer: Thank you very much for all your comments. We are very pleased that our work is interesting to you.

Comment 1: Introduction; the introduction and entire article have an excellent flow of background context and relevant references. From the technical point of view, the paper provides a straightforward approach and the scientific experiment seemed to be sound in nature using inertial sensors and real-time heart measurement sensor output parameters to make judgements. NASA and other similar organisations have confirmed trampolining and the positive effects of g-force as an excellent form of exercise. It is suggested that the references in line 29 also include the following trampolining reference: Eager D., Chapman C., Bondoc K., Characterisation of trampoline bounce using acceleration, 7th Australasian Congress on Applied Mechanics ACAM7, Adelaide, Australia, 9-12 December 2012.

Answer: Thank you for your suggestion. The indicated item has been included in the bibliography.

Comment 2: Introduction; a brief background on unexplanatory heart failures in young adults with good exercise history ought to be included in the introduction.

Answer: Appropriate information has been included in the revised manuscript.

Comment 3: Discussion; the discussion should refer to the individual figure and table data in the sentences where applicable.

Answer: The required information has been completed.

Comment 4: Abstract; in the abstract please include how the integration of different components in the SMART system contributed to the exercise tolerance assessment besides HR loading (Polar H10 chest strap) and its effectiveness. This can include the execution and protocols used from the SMART system factors/elements to arrive at the results.

Answer:

Comment 5: Methods; in lines 124-125, the point "mean heart rate from SMART and ECG measurements" is ambiguous, please rephrase/elaborate.

Answer: According to your suggestion, the point has been clarified

Comment on the Quality of English Language: The Patent and funding sections at the rear of the paper require editorial correction.

Answer: The text has been checked again for linguistic correctness. The corrections have been marked in red.

Reviewer 3 Report

Comments and Suggestions for Authors

This manuscript is well designed and written. The objective and means are clear. And the description and statistics are abundant. But I have a small issue, the placement of the SMART system is not invovled, and I think its placement will influence the results, please supplement some description, figures, or discussion in the revised manuscript.

Author Response

General comment: This manuscript is well designed and written. The objective and means are clear. And the description and statistics are abundant. But I have a small issue, the placement of the SMART system is not invovled, and I think its placement will influence the results, please supplement some description, figures, or discussion in the revised manuscript.

Answer: Thank you very much for your comment. Depending on measurement needs, IMU sensors can be placed in various places (sacrum, C7 vertebra, lower leg). For the purposes of this study, sensors placed on the lower legs were used to determine the number of steps. Appropriate information has been included in the revised manuscript.

Round 2

Reviewer 1 Report

Comments and Suggestions for Authors

Thank you to the authors for making the suggested changes.

Please note that comment 12 has not been responded to and dealt with. This is in relation to lines 253 and 254 in the revised manuscript.

Comments on the Quality of English Language

The use of English is rather good but needs to be checked.

Author Response

Thank you, once more time for giving us the opportunity to submit a revised draft of the manuscript “SMART system in the assessment of exercise tolerance in adults”.  We appreciate the time and effort that you and the reviewers dedicated to providing feedback on our manuscript and are grateful for the insightful comments on and valuable improvements to our paper.

We hope that the manuscript, after careful corrections, will meet your high standards.

The answers are given below point by point.

All modifications in the manuscript are highlighted in red.

Comment 12: Discussion; lines 228-229: It says no relationship was found between age and morphological parameters and the estimated level of fitness. This cannot be said as there was no statical analysis conducted to determine the relationship. If a relationship is to be included, then the appropriate statistical test needs to be used. Furthermore, this would be an additional aim to the study.

Answer: Thank you very much for this comment and we apologize for missing this reply during the first review.

Statistical analysis did not show any differences between age and morphological parameters and the level of fitness within the research groups. The assessment of the correlation between somatotype and the level of performance of the examined people is the topic of our next article.

This information has been added to the article.

Comment on the Quality of English Language: The quality of English is good but needs a careful read to make the text clearer.

Answer:

Linguistic corrections were added to the text.
